# A Peer-Based Educational Intervention Effects on SARS-CoV-2 Knowledge and Attitudes among Polish High-School Students

**DOI:** 10.3390/ijerph182212183

**Published:** 2021-11-20

**Authors:** Maria Ganczak, Oskar Pasek, Łukasz Duda-Duma, Julia Komorzycka, Karol Nowak, Marcin Korzeń

**Affiliations:** 1Department of Infectious Diseases, University of Zielona Gora, 65-417 Zielona Gora, Poland; 2Student Research Group, University of Zielona Gora, 65-417 Zielona Gora, Poland; oskar.pasek2@gmail.com (O.P.); l.duda.duma@gmail.com (Ł.D.-D.); jmkomorzycka@gmail.com (J.K.); nowakkarol80@gmail.com (K.N.); 3Department of Methods of Artificial Intelligence and Applied Mathematics, West Pomeranian Institute of Technology, 71-210 Szczecin, Poland; mkorzen@wi.zut.edu.pl

**Keywords:** SARS-CoV-2, knowledge, attitudes, adolescents, intervention

## Abstract

To support high school students to develop knowledge they need to adhere to control measures during the pandemic, a peer-based educational intervention on SARS-CoV-2 was developed and its impact was evaluated. Multistage random sampling was used. The 50 min peer-based intervention was conducted by final year medical students. Baseline and post-intervention knowledge and attitudes were assessed. Significance was tested by McNemar’s/Wilcoxon rank tests. Of 518 participants (mean age 17.8 years ± 0.43), 81.0% did not receive any school-based education on SARS-CoV-2. After intervention, the knowledge score improved from 65.2% to 81.6%, attitudes from 63.2% to 70.8% (both *p* < 0.0001). The effect size after the intervention compared to pre-intervention showed moderate improvement of knowledge, but not attitudes (d = 0.46 and d = 0.18, respectively). Pre- and post-intervention, females, students in non-science programs, living in cities < 250,000 inhabitants had lower knowledge, while fewer males, non-science program students, living in smaller cities presented positive attitudes. Before intervention, 67.0% students correctly named SARS-CoV-2 preventive methods and 73.6% were concerned COVID-19 is a serious disease; these improved after intervention (to 80.1% and 86.3%; *p* < 0.0001). The intervention was not very successful in increasing the intent to vaccinate for COVID-19 (pre-intervention 52.9%, post-intervention 56.4%; *p* < 0.007). Peer-based teaching for high school students can be effective in increasing SARS-CoV-2 knowledge and awareness. More efforts are needed to improve attitudes and enhance acceptance of vaccination against COVID-19.

## 1. Introduction

Despite available vaccines, the world’s battle against COVID-19 continues [1,2]. Although adolescents are reportedly at lower risk of severe disease and death than persons in other age groups, they can experience infection [3,4]. Even though some infections among asymptomatic adolescents are likely to have gone undetected, they may fuel the spread of SARS-CoV-2 across generations and affect those at higher risk of severe illness [3,4,5]. In addition, some adolescents can also suffer from serious illness even if they are at lower risk of severe COVID-19. Moreover, asymptomatic or mild cases can result in sequelae, such as myocardial inflammation [6]. 

Young people’s adherence to control measures, such as using face masks, social distancing, and regular hand washing, is crucial to reduce SARS-CoV-2 transmission among their contacts and communities [4]. However, previous reports identified young adults as being less likely than other age groups to adhere to COVID-19 prevention measures, including immunization [3,7,8]. To improve adherence, adolescents should be armed with adequate knowledge about SARS-CoV-2 infection. This in turn may have a positive effect on their attitudes and practices.

High school years are crucial for acquiring adequate knowledge and utilizing reliable information sources, which may play an important role in the adoption of positive health behaviours, including acceptance of vaccinations and health decision-making practices [9,10]. As colleagues are often regarded as a more credible and non-judgmental source of information, in contrast to authoritative adult gatekeepers and stakeholders, young people tend to discuss most issues with their peers [11]. Therefore, youth peer education (YPE) is used increasingly as a health promotion strategy to reach and sensitize young people to health-related issues [12,13]. Several studies demonstrated gains in knowledge and changes in attitude in school-based HIV/AIDS YPE programs [14,15,16,17,18]. However, research on YPE programs’ effectiveness regarding COVID-19 is scarce. Given the significant role of the young in SARS-CoV-2 transmission and poor adherence to prevention measures, there is an urgent need to measure whether YPE programs make a valuable contribution to knowledge and attitude change. 

Medical students cannot continue uninvolved in the battle against COVID-19. However, their assistance should be rather restricted to outside hospital wards [19]. For instance, they could volunteer to deliver up-to-date information regarding SARS-CoV-2, setting themselves as role models by adopting preventive behaviours and urging others to follow these measures [19,20,21].

In the above-mentioned context, the study objective was to train medical students in SARS-CoV-2 related issues, then develop a peer-based educational intervention among high school students, followed by an evaluation of its impact. It was hoped that by improving knowledge, the intervention would alter misconceptions regarding COVID-19 and thus help with disease control. 

## 2. Materials and Methods

### 2.1. Design and Setting

A before–after school-based analytic survey with pre-intervention and immediate post-intervention phases was designed. This choice was based on our previous experience gained during an HIV/AIDS educational intervention [14]. 

The territory of Poland is divided into 16 provinces. Multistage random sampling selected 5 provinces with 5 capital cities (Zielona Góra, Poznań, Toruń, Opole, Lublin; population sizes range between 100,000 and 500,000 inhabitants), then 2 high schools per city were randomly selected; at each selected school, half in final year classes in a life science program (including extra hours per week oriented to biology and chemistry) and half in a non-science program were selected in the last step (25 classes total), with the assumption that life science students were more acquainted with biology and general health principles. 

### 2.2. Study Population and Sampling

Based on census data, there were about 601,700 students at Polish high schools at the beginning of the 2020–2021 academic year and 62.7% of these were female [22]. The field survey was carried out in September 2020. The study population consisted of final year high school students aged from 17 to 19 years. Respondents were informed that participation was anonymous and voluntary. The sample size was calculated by a sample size calculator based on the reference population of final year high school students, with a 95% confidence level and a 5% margin error, as previously reported [23]. A sample of 383 was sufficient to detect, with a power of 80% and a significance level of 5%, the absolute difference in positive response to knowledge and attitude questions between subgroups, e.g., males and females, of 15%. This sample size was also enough to demonstrate with the same power and significance level an absolute improvement after intervention of 15% in the fraction of participants, with good knowledge or attitude, assuming that with at most 80% of participants, the intervention would not change knowledge or attitudes. 

### 2.3. Data Collection 

A structured self-administered anonymous questionnaire was used as the data collection instrument. It was developed by the authors after literature review [23,24,25,26]; the opinions of a panel of 4 experts composed of an infectious disease specialist, an epidemiologist, an expert in family medicine, and an expert in public health were also considered. A pilot study was conducted on a group of 20 students; results were included in the study. After reviewing their comments, amendments were made to improve the clarity of the questions. The pre-intervention section of the questionnaire was structured in parts:Demographics: age, gender, division, and facility location.General SARS-CoV-2/COVID-19 knowledge: epidemiology, transmission, clinical course, treatment, and preventability (16 items rated as true/false/don’t know).Attitudes regarding SARS-CoV-2/COVID-19, such as “I am concerned COVID-19 is a serious disease” (10 items rated as true/false/don’t know).

The post-intervention section included only the second and third part. Each section took about 7–10 min to complete, with the questionnaires being completed in classrooms in the presence of the research team members, after explaining the purpose of the study and obtaining written consent. High school students were asked to answer the questionnaire before and immediately after the intervention. This procedural design excluded the possibility that the improvement in knowledge was a result of the other information that the students were prompted to seek after participating in the intervention.

### 2.4. Selection, Training, and Curriculum for Student Peer Educators

In March 2020, the University of Zielona Gora Medical Faculty students who were the best in their class, including the highest grades obtained during an Infectious Diseases course, were invited to join the Student Research Group. This was led by an academic professor specializing in infectious diseases and epidemiology. During the following months, their teaching skills, including a listener-centred approach and effective communication, as well as personal qualities, such as interpersonal relationships and enthusiasm for teaching—all essential to effective role modelling—were examined. Finally, 6 fifth year medical students, 3 females and 3 males aged 22–24 years, were chosen to conduct peer-based educational interventions at the selected high schools. 

Due to epidemiological reasons, student training was carried out online, with the use of Discord software. A concise curriculum was developed. This included:Sampling (provinces, schools, classes); writing a draft of the project and an application to the Bioethics Committee; and preparing letters to school headmasters—5 h.SARS-CoV-2/COVID-19 teaching methods: a PowerPoint presentation vs. a workshop and face-to-face teaching—3 h.Work on SARS-CoV-2/COVID-19 knowledge seminar—5 h.Short SARS-CoV-2/COVID-19 attitudes workshop: types of mini-workshops on beliefs and attitudes and discussion of their potential impact on students in their perception of personal risk and vulnerability—2 h.Project work to review and revise a questionnaire and factual presentation on SARS-CoV-2/COVID-19—5 h.

### 2.5. Intervention

Due to COVID-19 pandemic travel restrictions, students were assigned to provinces and schools that were located closest to their places of residence.

The intervention consisted of a PowerPoint presentation about SARS-CoV-2/COVID-19 (40 min) and a short participatory mini-workshop on attitudes, based on master imagery (10 min). The presentation included: epidemiology, modes of transmission, clinical course, treatment, and prevention. 

In September 2020, when the educational intervention was conducted, there was no COVID-19 vaccine available for the general population. Therefore, the presentation focused on those vaccine candidates who were under investigation regarding various phases of randomized clinical trials. Additionally, the need for a COVID-19 vaccination was also stressed during the presentation, as well as the benefits of vaccination at the individual and the general population level. 

The objectives of the COVID-19 teaching were: to increase general knowledge about SARS-CoV-2/COVID-19; to clarify transmission modes, risk factors, and preventive methods; to reduce existing myths and unconfirmed information regarding SARS-CoV-2 infection; and to increase SARS-CoV-2 awareness and improve school student attitudes toward SARS-CoV-2/COVID-19, including the willingness to vaccinate.

### 2.6. Knowledge and Attitude Scoring

For knowledge questions, each correct answer was given 1 point. A positive result included a change to a correct response or to unknown from an incorrect response for yes/no/don’t know knowledge questions. For 10 attitude questions, which used a 3-point scale with response categories ranging from “agree” to “disagree” and “don’t know”, 3 points were given for the most positive attitude and 0 for the most negative (total of 0–30 points).

### 2.7. Data Analysis

Data were converted to a Microsoft Excel spreadsheet, then analysed using Statistica PL version 12.5 (StatSoft Inc., 2016, Kraków, Poland) and a statistical software package (R Foundation for Statistical Computing, Vienna, Austria) [27]. A descriptive analysis was performed that considered demographic characteristics and answers provided by the sample as a whole and grouped on the basis of: 1. gender; 2. type of division: life science/others; and 3. facility (school) location (towns of ≥250,000/<250,000 inhabitants). Continuous outcomes were expressed as mean values ± standard deviation (SD). For dichotomous knowledge data, baseline and post-intervention levels of knowledge were compared by gender, division, and school location for individual questions by means of McNemar’s test. For ordinal attitude data, a Wilcoxon signed-rank test was used. The differences between pre- and post-intervention scores were estimated overall and stratified for gender, division, and school location. Effect sizes (Cohen’s d) were calculated to determine the magnitude of the differences between pre- and post-intervention time points with 0.2 considered a small ES, 0.5 as a moderate ES, and ≥0.8 as a large ES [28,29].

## 3. Results

Out of 10 schools invited, 1 refused due to understaffing caused by numerous absences related to SARS-CoV-2 infections and subsequent isolation/quarantine. The total number of students who agreed to participate was 518 (response rate 100%). Mean respondent age was 17.8 ± 0.43 (range 17–19 years); 21.4% (*n* = 111) were 17 years old, 77.4% (*n* = 401) were 18 years old, and 1.2% (*n* = 6) were 19 years old; 66% were females; and 72.6% were in the life science program. Almost three quarters (72.6%) attended schools located in towns with <250,000 inhabitants. The vast majority of students (81%) had received no previous teaching relating to SARS-CoV-2/COVID-19.

### 3.1. Knowledge Scores

The combined mean knowledge scores among students improved from 9.13 (65.2%) to 11.43 (81.6%) after the intervention (with a moderate effect of ES = 0.46), *p* < 0.0001, meaning a relative increase of 16.4% (Table 1). Females had a lower baseline and post-intervention scores than males. Both genders recorded a significant (*p* < 0.0001) improvement in knowledge level, with a moderate effect (ES = 0.56) observed in females and a low effect (ES = 0.36) observed in males. In females, knowledge scores increased from 8.74 (62.4%) to 11.31 (80.8%), a relative increase of 18.4%, whereas for males, it was a relative increase of 15.5% from 9.33 (66.6%) to 11.49 (82.1%). 

Life science program students demonstrated greater knowledge before and after the intervention: 9.47 (67.6%) and 11.74 (83.9%), respectively, a relative increase of 16.3%, with a moderate effect (*p* < 0.0001, ES = 0.59). However, students in other programs recorded a greater increase in knowledge: 8.22 (58.7%) to 10.62 (75.9%), an increase of 17.2%, (*p* < 0.0001, ES = 0.34). 

Before and after the intervention, participants from towns with ≥250,000 inhabitants demonstrated greater knowledge than students from schools in the smaller cities, an increase from 9.82 (70.1%) to 12.06 (86.1%) and 8.87 (63.4%) to 11.20 (80.0%), respectively. There was a significant increase in knowledge scores (16.0% and 16.6%, respectively, both *p* < 0.0001) with a large effect (ES = 0.96) on students from larger cities and a low effect (ES = 0.41) on those living in smaller towns.

Pre-intervention knowledge ranged from about 5 to 8% on questions relating to potential therapeutic options for COVID-19, between 25% and 40% on the SARS-CoV-2 epidemiological situation, and between 38 and 46% on SARS-CoV-2 origin. Students scored better regarding general knowledge questions (82–97%), COVID-19 symptoms (70–94%), and prevention (67%) (Appendix A). Post-intervention, there were significant changes on the vast majority of questions.

### 3.2. COVID-19 Attitude Scores Pre- and Post-Intervention

The mean attitude score for all students combined improved after intervention from 6.92 (69.2%) to 7.81 (78.1%), a relative increase in scores of 8.9% by an ES = 0.18 (*p* < 0.0001) (Table 2). A significant increase was observed in both genders (*p* < 0.0001): in females, the increase was from 6.97 (69.7%) to 7.88 (78.8%), a relative increase of 9.1% (*p* < 0.0001), and in males, from 6.84 (68.4%) to 7.66 (76.6%), an increase of 8.2%; an increase in scores by an ES of 0.15 and ES of 0.20, respectively. 

Pre- and post-intervention life science program students outperformed other student divisions in attitude scores: 7.22 (72.2%) and 8.14 (81.4%), a relative increase of 9.2%, vs. 6.15 (61.5%) and 6.92 (69.2%), a relative increase of 7.7%, respectively. Improvements in attitude scores were significant (both *p* < 0.0001); however, with low effects (calculated ES = 0.22 and ES = 0.12, respectively). High school students living in towns with ≥250,000 inhabitants outperformed those living in smaller cities: 7.75 (77.5%) and 8.77 (87.7%), a relative increase of 10.2%, vs. 6.61 (66.1%) and 7.45 (74.5%), a relative increase of 8.4%, respectively. Attitude scores improved significantly (both *p* < 0.0001); however, with a low effect (ES = 0.15) of participants living in smaller cities and a moderate effect (ES = 0.47) of those living in bigger towns.

Table 3 presents correct answer frequency in attitudes toward SARS-CoV-2/COVID-19 pre- and post-intervention by gender, curriculum track, and school location. There were differences between genders regarding all attitude statements before and after intervention. For instance, before the intervention, fewer (69%) males than females (75%) believed that COVID-19 is a serious disease; the frequency increased significantly after the intervention. Only 17% of males and 10% of females before the intervention agreed that COVID-19 does not concern them. After the intervention, this actually dropped in males, but not in females. Before the intervention, more females than males were concerned that their parents (49% vs. 37%) and grandparents (61% vs. 53%) will contract COVID-19; after the intervention, this increased significantly in both genders; however, male attitude scores still remained at a relatively low level. Only 60% percent of males and 49% of females declared that they would vaccinate against SARS-CoV-2 if a vaccine was available; after the intervention, this increased slightly, however significantly. Students expressed positive attitudes toward SARS-CoV-2 preventive measures: 80% males and 78% females thought it is important to use face masks in crowded places, and the vast majority stated it is important to wash their hands frequently (females 97%, males 90%); this increased in both genders after the intervention. 

As illustrated in Table 3, similar differences regarding the basic and post-intervention scores were observed between divisions regarding all attitude statements, with students representing non-science programs scoring less than those in life science programs. 

Students living in towns with ≥250,000 inhabitants presented more positive attitudes, before and after the intervention, when compared to those living in smaller cities. Participants from the schools located in the bigger towns presented more appropriate attitudes toward preventive methods: 90% agreed that it is important to wear a mask in crowded places vs. 74.5% from smaller cities. After the intervention, the scores improved to 98% and 86%, respectively.

## 4. Discussion

### 4.1. Results Overview

To the best of our knowledge, this was the first study that assessed the effects of a short peer-based educational intervention on SARS-CoV-2 knowledge and attitudes among high school students. Other cross-sectional studies also concentrated on assessing knowledge, attitudes, and practices (KAP) in different population groups, however, they did not evaluate the effects of an educational intervention regarding KAP. Before intervention, final year high school student knowledge regarding SARS-CoV-2 was inadequate and their attitudes inaccurate. The effect size after the intervention compared to pre-intervention showed moderate improvement of knowledge, but not attitudes (d = 0.46 and d = 0.18, respectively).

### 4.2. Effects of a Short Peer-Based Educational Intervention 

The lesson learned from the previous SARS outbreak (2003) suggests that knowledge and attitudes toward an infectious disease may be associated with the level of panic in the community; this, in turn, can make attempts to prevent the spread of the infection difficult and more complicated [30,31]. Effective education may boost a country’s response to a pandemic. 

Most of our participants were poorly knowledgeable about SARS-CoV-2 with an overall correct rate of 58% after a knowledge test. The basic score before intervention was lower than that reported in other recently published studies [25,32,33]. However, the above-mentioned studies assessed knowledge using online surveys rather than filling out questionnaires in the presence of research team members; this means that some external sources of information could have been used.

According to the study results, pre- and post-intervention, females were less knowledgeable than males. This was in line with the results of some previous surveys published in Asian developing countries [34,35]. However, other authors reported better knowledge among female participants [25,26,32,36,37]. Further studies are needed to better asses this issue.

Additionally, pre- and post-intervention, adolescents living in bigger cities and studying in a life science program were found to have better knowledge about SARS-CoV-2. This was consistent with other findings [23,32,36,37] and may be attributed to student reliance on digital sources of information with easier access in urban settings, as well as higher levels of health-oriented education in life science courses. For example, students with education based around biology were found to have more advanced knowledge about viruses, vaccines, and drug targets in the studies conducted in Italy and Japan [23,37]. 

Before intervention, only two thirds of students correctly named basic preventive methods regarding SARS-CoV-2 infection. This improved significantly after the intervention and confirmed young people are now better equipped to protect themselves and others from getting infected. Some other gaps in the understanding of COVID-19 were also uncovered. Generally, adolescents’ knowledge was oriented mainly to practical issues, such as transmission routes and preventive methods. 

We were able to demonstrate that our educational intervention significantly improved attitudes related to COVID-19. For instance, almost three quarters of participants were concerned that COVID-19 is a serious disease, which increased after the intervention. The attitudes towards taking precautions to prevent COVID-19 also improved: before intervention 79% of students stated that it is important to use face masks in public spaces, this increased to 89% after intervention. 

However, while the *p*-value showed that the intervention had an impact on attitude scores, the size of the impact was low. This might be due to the fact that the study was mainly based on a knowledge intervention, accompanied by a very short, 10 min workshop. Other studies showed that information together with master imagery and skill-building workshops were more effective in raising knowledge levels and changing attitudes than compared to information alone [14,38]. Although we did not study the knowledge intervention exclusively, it was shown in HIV/AIDS educational interventions that information was not sufficient to change attitudes, and programs were most effective when they provide both information and behavioural skills [38,39]. Further research is urgently needed on the effectiveness of SARS-CoV-2/COVID-19 educational interventions regarding attitude change.

Of note, students living in bigger cities and studying in a life science program began and finished at higher attitude scores. This was in accordance with other studies that showed an association between attending life science courses and healthy behaviours [20,40,41]. Moreover, the effect of the intervention was moderate from participants attending schools in bigger cities and low in students from the small towns.

Nevertheless, much remains to be done, especially regarding COVID-19 vaccinations: the intervention was not particularly successful in increasing the intention to vaccinate. This was of no surprise; younger individuals may believe that COVID-19 poses a less serious threat to themselves than to other age groups [42]. Additionally, several studies reported that East European countries had the lowest intention to vaccinate among their young populations [10,43,44,45]. For instance, a survey conducted in June 2020 showed that only 43% of Poles between 18 and 25 years of age wanted to be vaccinated against SARS-CoV-2 if the vaccine became available [43]. This percentage was lower than the figure reported in our study (55%) carried out three months later. Alarmingly, more than one third (37%) of the reluctant young respondents indicated that their minds could be neither changed by information regarding vaccine safety and efficacy delivered by a family doctor or another expert, nor being threatened with potential hefty fines. In another Polish study, conducted at the beginning of 2021, the authors presented respondents with different sets of information relating to the COVID-19 vaccination [44]. After reading the information package, they indicated whether they would be willing to be vaccinated or not; only 46.5% of the participants were willing to do so. Furthermore, none of the COVID-19-related messages used were effective in reducing vaccine hesitancy. The above-mentioned studies were consistent with the results presented here and suggest that in Poland, regardless of changes in the intensity of the pandemic, the percentage of people who are sceptical regarding vaccination remains high and—in many cases—their minds would remain unchanged by any newly delivered information. These might be factors impacting the partial failure of our educational intervention with regard to COVID-19 vaccination willingness. 

Interestingly, we found large gender differences pre- and post-intervention, with females starting and finishing with much lower intent, but showing better improvement after intervention. However, even after the intervention, the intent to vaccinate for COVID-19 among females remained at an alarmingly low level. While more males than females opted for being immunized pre-intervention, this did not improve much after intervention.

Although, overall, our intervention was successful in increasing knowledge scores and to some extent improving attitudes, gaps between genders, divisions, and place of residence remained. Therefore, regarding future educational campaigns, particular attention should be focused on females, students living in less urbanized environments, and those studying in non-science programs.

### 4.3. Peer-Based Education

The basis of many behavioural change interventions is that people who are similar to the target population will be the most effective change agents, due to their empathy and cultural understanding [9]. Therefore, peer-to-peer programs are widely used among young people. Some studies showed that such programs can have an impact on young people’s KAP [9,41]. As reported by others, education provided by medical students might be more effective in increasing knowledge and changing attitudes compared with other university students, because there may have been a higher level of trust in their expertise [14]. 

Many authors recommend that medical students should volunteer in the COVID-19 pandemic [19,20,21]. And indeed, the enthusiasm and interest of medical students and their empathy and trust among high school students cannot be undervalued in the context of positively influencing the educational process and changing attitudes. As reported by other authors, education provided by medical students might be more effective in improving knowledge compared to teachers, because there may have been a higher level of trust in their expertise [14]. For instance, several programs for HIV/AIDS prevention that include school education and medical student involvement, were positively evaluated for effectiveness in changing targeted knowledge, attitudes, and behaviours [14,46,47]. Of note, before educational interventions, the final year medical students from our university were all given the same extended professional training during their curriculum. Furthermore, they completed an extra 20 h of training strictly oriented to the SARS-CoV-2/COVID intervention. Secondly, their particular ability to present less-stereotypical beliefs than school teachers cannot be overestimated [47]. They are young, and as such, were able to communicate more easily with the high school students. Their speaking “the same language” could facilitate the transfer of information. Another advantage was that such peer educators could have been seen, not as an authority telling them how to behave, but as another—although more knowledgeable—member of their own group [48]. Another issue in favour of training this category of peer educators is that many school teachers find it difficult to discuss topics, such as SARS-CoV-2/COVID-19, with adolescents because they have not undergone medical training. 

It is also important that peer education was reported to have positive outcomes for peer educators in their future professional work [14]. Our junior colleagues undoubtedly stated that by joining and taking part in an educational intervention in high schools, they improved their own SARS-CoV-2 knowledge. Furthermore, this inspired them to be vaccinated against COVID-19 (in spring 2021) and to actively join an online National Health Fund campaign (in June 2021) promoting COVID-19 immunization among young people.

### 4.4. Limitations

The main limitation was that the study did not include a control group; the participating students were their own controls. After discussing the pros and cons of such an approach, the authors concluded that, due to the ongoing epidemic, it was problematic to explain to the headteachers and students that, serving as control groups, they would not receive any education regarding SARS-CoV-2. However, in view of the importance of SARS-CoV-2 prevention among young people, RCTs or other rigorous approaches to evaluate school-based interventions is recommended [14]. 

Secondly, although not assessed in this study due to different communication skills, some medical student educators might have performed more effectively than others. Moreover, as we did not assess the effects of the peer-based intervention conducted by medical students in comparison to an intervention conducted by school teachers, it was not possible to detect any possible differences between those two modes of intervention.

The study presented hypothetical situations, which did not ensure that students will be compassionate and/or present positive attitudes in real situations. Finally, due to the time frame of the study, the long-term effects of the intervention were not assessed. Further studies evaluating such effects would be of value.

## 5. Conclusions 

As evidenced by our results, peer education led by medical students may be effective in transmitting factual health information. Our research showed the importance of a simple before–after assessment prior to implementing educational programs related to a new topic and with different trainers, such as medical students. Such an assessment should be performed to guarantee that educators work efficiently to deliver messages properly, so they are correctly understood.

The results obtained were supportive regarding the effectiveness of educational SARS-CoV-2/COVID-19 interventions in reducing gaps in basic knowledge and increasing awareness among high school students. It seemed, however, that issues related to the low influence of our intervention on the improvement of attitudes require further research. 

More efforts are also needed to enhance the adolescents’ acceptance of vaccination. As boys were more reluctant to change attitudes toward COVID-19 immunization, while girls in general expressed much lower intent to vaccinate, different messages and educational approaches should be tailored to female and male students.

We recommend that educational programs, similar to the one described, be implemented in high schools, including both an informative presentation and attitude workshops, which would be crucial in helping to positively modify attitudes. In planning educational programs on a national level, the scientific trustworthiness and recognized status of SARS-CoV-2 educators should be thoroughly considered. With the lack of other health professionals, currently involved in COVID-19 epidemic management, medical students can provide a link between such professionals and a target group for intervention. Furthermore, in the upcoming months, the education provided can form a solution to help bridge gaps in the limited human resources required to provide effective education in schools, not only about SARS-CoV-2 related issues, but, more broadly, topics related to infection control and pandemic preparedness. Regarding any educational intervention, it remains crucial to constantly update the teaching portfolio and focus on evidence-based research that uses scientifically approved methods.

Of note, even if an intervention is successful in increasing awareness and knowledge and improving attitudes, it might not solely be effective enough in slowing SARS-CoV-2 transmission. There is no single factor limiting the number of new infections [49], but rather a concerted, thoroughly planned strategy that acts through numerous control measures. Each plays a different role, depending on the regional, provincial, or individual risk of exposure to SARS-CoV-2, due to the multiple potential modes of transmission.

## Figures and Tables

**Table 1 ijerph-18-12183-t001:** Overall knowledge scores pre- and post-COVID-19 intervention by selected variables; final year high school students, Poland, 2020 (*n* = 518).

Variable	*n*	Pre-Intervention	SD	Post-Intervention	SD	*p*
Total	518	9.13	2.28	11.43	2.18	<0.0001
Gender						
Males	342	8.74	2.69	11.31	2.64	<0.0001
Females	176	9.33	2.01	11.49	1.90	<0.0001
Division						
Life science	376	8.87	2.41	11.20	2.38	<0.0001
Other	142	9.82	1.69	12.06	1.35	<0.0001
School Location by Number of Inhabitants						
<250,000	376	9.47	2.04	11.74	1.88	<0.0001
≥250,000	142	8.22	2.62	10.62	2.67	<0.0001

**Table 2 ijerph-18-12183-t002:** Attitudes toward SARS-CoV-2 and COVID-19 pre- and post-intervention by gender, division, and school location; final year high school students, Poland, 2020 (*n* = 518).

Variable	*n*	Pre-Intervention	SD	Post-Intervention	SD	*p*
**Total**	518	6.92	2.28	7.81	2.15	<0.0001
Gender						
**Males**	176	6.84	2.39	7.66	2.32	<0.0001
**Females**	342	6.97	2.22	7.88	2.06	<0.0001
Division						
**Life science**	376	7.22	2.15	8.14	1.89	<0.0001
**Other**	142	6.15	2.44	6.92	2.54	<0.0001
School Location by Number of Inhabitants						
**<250,000**	376	6.61	2.42	7.45	2.29	<0.0001
**≥250,000**	142	7.75	1.58	8.77	1.35	<0.0001

**Table 3 ijerph-18-12183-t003:** Attitudes toward SARS-CoV-2 andCOVID-19 pre- and post-intervention by gender, school location, and curriculum track, final year high school students, Poland, 2020 (*n* = 518).

Statement	Agree		Disagree		Not Sure/I Don’t Know		Positive Shift	Negative Shift	*p*
	*n*	%	*n*	%	*n*	%			
COVID-19 Is a Serious Disease									
Males									
Before	123	69.9	35	19.9	18	10.2			
After	143	81.3	22	12.5	11	6.3			
Difference	20	16%	−13	−37%	−7	−39%	18%	4%	<0.0001
Females									
Before	258	75.4	48	14.0	36	10.5			
After	304	88.9	15	4.4	23	6.7			
Difference	46	18%	−33	−69%	−13	−36%	19%	5%	<0.0001
<250,000									
Before	269	74.1	69	19.0	25	6.9			
After	315	83.8	36	9.6	25	6.6			
Difference	46	13%	−33	−50%	0	−3%	19%	6%	<0.0001
≥250,000									
Before	112	78.9	14	9.9	16	11.3			
After	132	93.0	1	0.7	9	6.3			
Difference	20	18%	−13	−93%	−7	−44%	18%	0%	<0.0001
Life science									
Before	288	76.6	51	13.6	37	9.8			
After	339	90.2	17	4.5	20	5.3			
Difference	51	18%	−34	−67%	−17	−46%	18%	2%	<0.0001
Other									
Before	93	65.5	32	22.5	17	12.0			
After	108	76.1	20	14.1	14	9.9			
Difference	15	16%	−12	−38%	−3	−18%	20%	8%	0.002
COVID-19 Doesn’t Concern Me									
Males									
Before	30	17.0	125	71,0	21	11.9			
After	20	11.4	146	83.0	10	5.7			
Difference	−10	−33%	21	17%	−11	−52%	20%	8%	0.0001
Females									
Before	34	9.9	251	73.4	57	16.7			
After	32	9.4	285	83.3	25	7.3			
Difference	−2	−6%	34	14%	−32	−56%	20%	16%	<0.0001
<250,000									
Before	56	14.9	259	68.9	61	16.2			
After	46	12.2	301	80.1	29	7.7			
Difference	−10	−18%	42	16%	−32	−52%	22%	14%	<0.0001
≥250,000									
Before	8	5.6	117	8.4	17	12.0			
After	6	4.2	130	91.5	6	4.2			
Difference	−2	−25%	13	11%	−11	−65%	14%	12%	0.003
Life science									
Before	35	9.3	288	76.6	53	14.1			
After	30	8.0	324	86.2	22	5.9			
Difference	−5	−14%	36	13%	−31	−58%	31%	9%	<0.0001
Other									
Before	29	20.4	88	62.0	25	17.6			
After	22	15.5	107	75.4	13	9.2			
Difference	−7	−24%	19	22%	−12	−48%	53%	20%	0.001
I Am Concerned My Parents Will Contract COVID-19									
Males									
Before	65	36.9	85	48.3	26	14.8			
After	95	54.0	63	35.8	18	10.2			
Difference	30	46%	−22	−26%	−8	−31%	51%	3%	<0.0001
Females									
Before	167	48.8	129	37.7	46	13.5			
After	210	61.4	93	27.2	39	11.4			
Difference	43	26%	−36	−28%	−7	−15%	27%	1%	<0.0001
<250,000									
Before	155	41.2	165	43.9	56	14.9			
After	198	52.7	131	34.8	47	12.5			
Difference	43	28%	−34	−21%	−9	−16%	31%	2%	<0.0001
≥250,000									
Before	77	54.2	49	34.5	16	11.3			
After	107	75.4	25	17.6	10	7.0			
Difference	30	39%	−24	−49%	−6	−38%	39%	0%	<0.0001
Life science									
Before	183	48.7	139	37.0	54	14.4			
After	240	63.8	98	26,1	38	10.1			
Difference	57	31%	−41	−29%	−16	−30%	32%	1%	<0.0001
Other									
Before	49	34.5	75	52.8	18	12.7			
After	65	45.8	58	40.8	19	13.4			
Difference	16	33%	−17	−23%	1	6%	39%	3%	0.0007
I Am Concerned My Grandparents Will Contract COVID-19									
Males									
Before	93	52.8	66	37.5	17	9.7			
After	109	61.9	51	29.0	16	9.1			
Difference	16	17%	−15	−23%	−1	−6%	19%	2%	0.0004
Females									
Before	210	61.4	96	28.1	36	10.5			
After	241	70.5	73	21.3	28	8.2			
Difference	31	15%	−23	−24%	−8	−22%	17%	4%	<0.0001
<250,000									
Before	203	54.0	128	34.0	45	12.0			
After	237	63.0	99	26.3	40	10.6			
Difference	34	17%	−29	−23%	−5	−11%	20%	3%	<0.0001
≥250,000									
Before	100	70.4	34	23.9	8	5.6			
After	113	79.6	25	17.6	4	2.8			
Difference	13	13%	−9	−26%	−4	−50%	14%	2%	0.0009
Life science									
Before	238	63.3	102	27.1	36	9.6			
After	272	72.3	74	19.7	30	8.0			
Difference	34	14%	−28	−27%	−6	−17%	16%	4%	<0.0001
Other									
Before	65	45.8	60	42.3	17	12.0			
After	78	54.9	50	35.2	14	9.9			
Difference	13	20%	−10	−17%	−3	−18%	23%	3%	0.002
Reporting Suspected COVID-19 Cases to the Health Authorities Is Crucial to Combat the Epidemic									
Males									
Before	139	79.0	20	11.4	17	9.7			
After	150	85.2	15	8.5	11	6.3			
Difference	11	8%	−5	−25%	−6	−35%	12%	14%	<0.0001
Females									
Before	260	76.0	18	5.3	64	18.7			
After	300	87.7	11	3.2	31	9.1			
Difference	40	15%	−7	−39%	−33	−52%	17%	5%	0.02
<250,000									
Before	279	74.2	34	9.0	63	16.8			
After	315	83.8	25	6.6	36	9.6			
Difference	36	13%	−9	−26%	−27	−43%	16%	9%	<0.0001
≥250,000									
Before	120	84.5	4	2.8	18	12.7			
After	135	95.1	1	0.7	6	4.2			
Difference	15	13%	−3	−75%	−12	−67%	13%	0%	0.0001
Life science									
Before	302	80.3	22	5.9	52	13.8			
After	339	90.2	12	3.2	25	6.6			
Difference	37	12%	−10	−45%	−27	−52%	13%	4%	<0.0001
Other									
Before	97	68.3	16	11.3	29	20.4			
After	111	78.2	14	9.9	17	12.0			
Difference	14	14%	−2	−13%	−12	−41%	21%	13%	0.006
It Is Important to Use Face Masks in Crowded Places to Minimize the Risk of SARS-CoV-2 Infection									
Males									
Before	140	79.5	23	13.1	13	7.4			
After	151	85.8	18	10.2	7	4.0			
Difference	11	8%	−5	−22%	−6	−46%	11%	14%	0.02
Females									
Before	268	78.4	35	10.2	39	11.4			
After	310	90.6	17	5.0	15	4.4			
Difference	42	16%	−18	−51%	−24	−62%	18%	7%	<0.0001
<250,000									
Before	280	74.5	54	14.4	42	11.2			
After	322	85.6	34	9.0	20	5.3			
Difference	42	15%	−20	−37%	−22	−52%	19%	10%	<0.0001
≥250,000									
Before	128	90.1	4	2.8	10	7.0			
After	139	97.9	1	0.7	2	1.4			
Difference	11	9%	−3	−75%	−8	−80%	9%	0%	0.001
Life science									
Before	310	82.4	30	8.0	36	9.6			
After	349	92.8	16	4.3	11	2.9			
Difference	39	13%	−14	−47%	−25	−69%	14%	8%	<0.0001
Other									
Before	98	69.0	28	19.7	16	11.3			
After	112	78.9	19	13.4	11	7.7			
Difference	14	14%	−9	−32%	−5	−31%	19%	11%	0.004
It Is Important to Wash Our Hands Often to Prevent the SARS-CoV-2 Infection									
Males									
Before	158	89.8	10	5.7	8	4.5			
After	161	91.5	11	6.3	4	2.3			
Difference	3	2%	1	10%	−4	−50%	6%	33%	0.46
Females									
Before	330	96.5	4	1.2	8	2.3			
After	335	98.0	0	0.0	7	2.0			
Difference	5	2%	−4	−100%	−1	−13%	2%	25%	0.15
<250,000									
Before	349	92.8	12	3.2	15	4.0			
After	357	94.9	10	2.7	9	2.4			
Difference	8	2%	−2	−17%	−6	−40%	5%	30%	0.11
≥250,000									
Before	139	97.9	2	1.4	1	0.7			
After	139	97.9	1	0.7	2	1.4			
Difference	0	0%	−1	−50%	1	100%	1%	33%	0.64
Life science									
Before	357	94.9	8	2.1	11	2.9			
After	364	96.8	6	1.6	6	1.6			
Difference	7	2%	−2	−25%	−5	−45%	3%	21%	0.08
Other									
Before	131	92.3	6	4.2	5	3.5			
After	132	93.0	5	3.5	5	3.5			
Difference	1	1%	−1	−17%	0	0%	5%	45%	0.80
COVID-19 Is a Preventable Disease									
Males									
Before	114	64.8	26	14.8	36	20.5			
After	128	72.7	17	9.7	31	17.6			
Difference	14	12%	−9	−35%	−5	−14%	25%	24%	0.04
Females									
Before	188	55.0	54	15.8	100	29.2			
After	228	66.7	33	9.6	81	23.7			
Difference	40	21%	−21	−39%	−19	−19%	36%	18%	<0.0001
<250,000									
Before	227	60.4	57	15.2	92	24.5			
After	250	66.5	42	11.2	84	22.3			
Difference	23	10%	−15	−26%	−8	−9%	26%	25%	0.02
≥250,000									
Before	75	52.8	23	16.2	44	31.0			
After	106	74.6	8	5.6	28	19.7			
Difference	31	41%	−15	−65%	−16	−36%	49%	9%	<0.0001
Life science									
Before	217	57.7	54	14.4	105	27.9			
After	262	69.7	30	8.0	84	22.3			
Difference	45	21%	−24	−44%	−21	−20%	33%	17%	<0.0001
Other									
Before	85	59.9	26	18.3	31	21.8			
After	94	66.2	20	14.1	28	19.7			
Difference	9	11%	−6	−23%	−3	−10%	29%	28%	0.16
Health Education Has a Significant Impact on COVID-19 Prevention and Control									
Males									
Before	140	79.5	13	7.4	23	13.1			
After	156	88.6	8	4.5	12	6.8			
Difference	16	11%	−5	−38%	−11	−48%	14%	11%	0.001
Females									
Before	284	83.0	24	7.0	34	9.9			
After	301	88.0	10	2.9	31	9.1			
Difference	17	6%	−14	−58%	−3	−9%	9%	16%	0.004
<250,000									
Before	290	77.1	36	9.6	50	13.3			
After	320	85.1	17	4.5	39	10.4			
Difference	30	10%	−19	−53%	−11	−22%	14%	14%	<0.0001
≥250,000									
Before	134	94,4	1	0,7	7	4,9			
After	137	96,5	1	0,7	4	2,8			
Difference	3	2%	0	0%	−3	−43%	3%	13%	0.23
Life science									
Before	323	85.9	18	4.8	35	9.3			
After	345	91.8	8	2.1	23	6.1			
Difference	22	7%	−10	−56%	−12	−34%	8%	9%	0.0001
Other									
Before	101	71.1	19	13.4	22	15.5			
After	112	78.9	10	7.0	20	14.1			
Difference	11	11%	−9	−47%	−2	−9%	19%	20%	0.03
I Would Vaccinate Myself against SARS-CoV-2 if a Vaccine Was Available									
Males									
Before	106	60.2	32	18.2	38	21.6			
After	110	62.5	29	16.5	37	21.0			
Difference	4	4%	−3	−9%	−1	−3%	11%	11%	0.38
Females									
Before	168	49.1	77	22.5	97	28.4			
After	182	53.2	64	18.7	96	28.1			
Difference	14	8%	−13	−17%	−1	−1%	11%	3%	0.004
<250.000									
Before	176	46.8	95	25.3	105	27.9			
After	185	49.2	85	22.6	106	28.2			
Difference	9	5%	−10	−11%	1	1%	13%	7%	0.13
≥250.000									
Before	98	69.0	14	9.9	30	21.1			
After	107	75.4	8	5.6	27	19.0			
Difference	9	9%	−6	−43%	−3	−10%	9%	0%	0.003
Life science									
Before	208	55.3	66	17.6	102	27.1			
After	228	60.6	53	14.1	95	25.3			
Difference	20	10%	−13	−20%	−7	−7%	12%	3%	0.0003
Other									
Before	66	46.5	43	30.3	33	23.2			
After	64	45.1	40	28.2	38	26.8			
Difference	−2	−3%	−3	−7%	5	15%	9%	11%	0.62

## Data Availability

The data underlying this article will be shared on reasonable request to the corresponding author.

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
