# Peer review of "A Peer-Based Educational Intervention Effects on SARS-CoV-2 Knowledge and Attitudes among Polish High-School Students"

_ijerph, 2021, doi:10.3390/ijerph182212183_

Round 1

Reviewer 1 Report

The study develops a peer-based educational intervention on SARS-Cov-2, adopted among high school students, and then evaluated its impact. After intervention, the knowledge score, perceived seriousness of COVID-19 and understanding of preventive methods were all improved. The authors also examined the impact of region, age, gender and life science program on the effect of the educational intervention. They found the intervention 22 was not much successful in increasing the intent to vaccinate for COVID-19.

This is an interesting study and very importantly practically amidst the pandemic. As authors suggested, this is the first study which assesses the effects of a short peer-based educational intervention on SARS-Cov-2 knowledge and attitudes among high school students. Involving medical students is also a very good idea in the fight against COVID-19. The research design and data are persuasive, though some procedural issues need to be clarified.

So, my first, and major concern about this study is both about its central validity and/or clarification of procedural issues. Did the author consider that the improvement in knowledge is a result of the other information that the students were prompted to seek after participating the intervention? If the procedural design has excluded this possibility (e.g. by immediately asking students to answer the questionnaire after the intervention), this is really a good study and practice. But if this has not been considered, we can nearly 100% be sure that the high school students may seek external information after the intervention. The more successful the intervention is, the more likely the high-schoolers may go to seek external information.

2. I am also concerned that what is the real impact of the peer-based educational intervention. If you have these high school students to have a whole class of COVID-19 education by their school teachers, their knowledge and perceived seriousness should also be improved. Then what is the difference between peer-based educational intervention and a regular class (if the latter is given)?

3. What is really the role played by the peers? The study is focused more on effect rather than the individual role played by peers in the actual process.

4. About vaccination intention, the explanation seems not enough. While young persons are those least likely to get vaccinated against COVID-19, this fact is not enough to analyze why the current peer-based educational intervention did not play strong enough role. Is it because the contents of the intervention hasn’t been focused on vaccination? Has the peer-based educational intervention tried to overcome misinformation? Studies reported that people in the East European countries have the lowest intention for vaccination (Feleszko, W., Lewulis, P., Czarnecki, A., & Waszkiewicz, P. (2021). Flattening the curve of covid-19 vaccine rejection—An international overview. Vaccines, 9(1). doi:10.1016/j.lanepe.2020.100012). Is this a factor impacting the partial failure of the intervention? Is it because peers may not particularly encourage vaccination because peers themselves are reluctant to get COVID-19 shot? All these need to be discussed and addressed, both in Discussion section and in method part when relevant.

Author Response

We would like to thank the Reviewer for all valuable and important comments.

  1. So, my first, and major concern about this study is both about its central validity and/or clarification of procedural issues. Did the author consider that the improvement in knowledge is a result of the other information that the students were prompted to seek after participating the intervention? If the procedural design has excluded this possibility (e.g. by immediately asking students to answer the questionnaire after the intervention), this is really a good study and practice. But if this has not been considered, we can nearly 100% be sure that the high school students may seek external information after the intervention. The more successful the intervention is, the more likely the high-schoolers may go to seek external information.

We would like to thank the Reviewer for this valuable comment.

Peer-educators asked students to answer the questionnaire immediately after the intervention. Therefore, the procedural design excluded the possibility that the improvement in knowledge was a result of the other information that the students were prompted to seek after participating the intervention.  The procedural design has been described in the paper as follows:

“The preintervention section of the questionnaire was structured in 3 parts (…). The postintervention section included only the second and third part. Each section took about 7-10 minutes to complete, with the questionnaires being completed in classrooms in the presence of research team members.”

Regarding the Reviewer’s comment, we described the whole process more precisely adding a relevant clarification as follows:

The preintervention section of the questionnaire was structured in 3 parts (…). The postintervention section included only the second and third part. Each section took about 7-10 minutes to complete, with the questionnaires being completed in classrooms in the presence of research team members. Students were asked to answer the questionnaire immediately after the intervention. This procedural design excluded the possibility that the improvement in knowledge was a result of the other information that the students were prompted to seek after participating the intervention.

  1. I am also concerned that what is the real impact of the peer-based educational intervention. If you have these high school students to have a whole class of COVID-19 education by their school teachers, their knowledge and perceived seriousness should also be improved. Then what is the difference between peer-based educational intervention and a regular class (if the latter is given)?
  2. What is really the role played by the peers? The study is focused more on effect rather than the individual role played by peers in the actual process.

We would like to thank the Reviewer for these important comments.

As we have not assessed the effects of the peer-based intervention conducted by medical students in comparison to an intervention conducted by school teachers, it is not possible to detect any possible difference between those two modes of intervention.  Of note, in Poland the content and quality of educational health-related school programs are varied and their impact is difficult to assess since they are provided by a range of professionals, mainly biology teachers. Regarding SARS-Cov-2/COVID-19, there has been no initial training planned for these professionals since the beginning of the pandemic. Therefore, the current content of the messages given to high-school students is difficult to summarize, and therefore evaluate.

We have amended the text according to the recommendations:

Limitations section

As we have not assessed the effects of the peer-based intervention conducted by medical students in comparison to an intervention conducted by school teachers, it is not possible to detect any possible difference between those two modes of intervention.  

Discussion section

As reported by other authors, education provided by medical students might be more effective in changing attitudes compared to teachers, because there may have been a higher level of trust in their expertise [Barss et al.]. For instance, several programs for HIV/AIDS prevention that include school education and medical student involvement, have been positively evaluated for effectiveness in changing targeted knowledge, attitudes, and behaviors [Barss et al., Sulak et al., Brettelle et al.]. Of note, before educational interventions final year medical students from our university were all given the same extended professional training during their curriculum. Furthermore, they completed an extra 20-hours of training strictly oriented to the SARS-Cov-2/COVID intervention.  Secondly, their particular ability to present less stereotyped beliefs than school teachers cannot be overestimated [Bretelle et al.]. They are young, and as such were able to communicate more easily with the high school students. The speaking of “the same language” could facilitate the transfer of information. Another advantage was that such peer educators could have been seen, not as an authority telling them how to behave, but as another - although more knowledgeable - member of their own group [Tolli].  Another issue in favour of training these kinds of peer educators is that many school teachers find it difficult to discuss topics such as SARS-Cov-2/COVID-19 with adolescents because they have not undergone medical training.

Barss, P.; Grivna, M.; Ganczak, M.; Bernsen, R.; Al-Maskari, F.; El Agab, H.; Al-Awadhi, F.; Al-Baloushi, H.; Al-Dhaheri, S.;  Al-Dhahri, J.; et al. Effects of a rapid peer-based HIV/AIDS educational intervention on knowledge and attitudes of high school students in a high-income Arab country. J. Acquir. Immune Defic. Syndr. 2009, 52(1), 86-98.

Sulak, P.; Sara, J.; MSa, H.; Dee Dee, A.; Fix, B.S.; Kuehl, T. Impact of an adolescent sex education program that was implemented by an academic medical center. Am. J. Obstet. Gynecol. 2006, 195, 78–84.

Bretelle, F.; Shojai, R.; Brunet, J.; Tardieu, S.; Manca, M. C.; Durant, J.; Ricciardi, C.; Boubli, L.; Leonetti, G. Medical students as sexual health peer educators: who benefits more? BMC Med. Educ. 2014, 14, 162.

Tolli, M. V. Effectiveness of peer education interventions for HIV prevention, adolescent pregnancy prevention and sexual health promotion for young people: a systematic review of European studies. Health Educ. Res. 2012, 27(5), 904–913.

  1. About vaccination intention, the explanation seems not enough. While young persons are those least likely to get vaccinated against COVID-19, this fact is not enough to analyze why the current peer-based educational intervention did not play strong enough role. Is it because the contents of the intervention hasn’t been focused on vaccination? Has the peer-based educational intervention tried to overcome misinformation? Studies reported that people in the East European countries have the lowest intention for vaccination (Feleszko, W., Lewulis, P., Czarnecki, A., & Waszkiewicz, P. (2021). Flattening the curve of covid-19 vaccine rejection—An international overview. Vaccines, 9(1). doi:10.1016/j.lanepe.2020. 100012). Is this a factor impacting the partial failure of the intervention? Is it because peers may not particularly encourage vaccination because peers themselves are reluctant to get COVID-19 shot? All these need to be discussed and addressed, both in Discussion section and in method part when relevant.

We would like to thank the Reviewer for these valuable comments. We addressed them as follows:

However, much remains to be done, especially regarding COVID-19 vaccinations: the intervention was not particularly successful in increasing the intention to vaccinate. This is of no surprise, younger individuals may believe that COVID-19 poses a less serious threat to themselves than to other age groups [38]. Additionally, several studies reported that East European countries have the lowest intention to vaccinate among their young populations [10,39-41]. For instance, a survey conducted in June 2020 showed that only 43% of Poles between 18-25 years of age wanted to be vaccinated against SARS-CoV-2 if the vaccine became available [39]. This percentage was lower than the figure reported in our study (55%) carried out three months later. Alarmingly, more than one third (37%) of the reluctant young respondents indicated that their minds could neither be changed by information regarding vaccine safety and efficacy delivered by a family doctor or another expert, nor being threatened with potential hefty fines. In another Polish study, conducted at the beginning of 2021, the authors presented respondents with different sets of information relating to the COVID-19 vaccination [40]. After reading the information package, they indicated whether they would be willing to be vaccinated or not; only 46.5% of the participants were willing to do so. Furthermore, none of the COVID-19 related messages used were effective in reducing vaccine hesitancy.  The above-mentioned studies are consistent with the results presented here and suggest that in Poland, regardless of changes in the intensity of the pandemic, the percentage of people who are skeptical regarding vaccination remains high and – in many cases - their minds would remain unchanged by any newly delivered information. These might be factors impacting the partial failure of our educational intervention with regard to COVID-19 vaccination willingness.

Feleszko, W.; Lewulis, P.; Czarnecki, A.; Waszkiewicz, P. Flattening the Curve of COVID-19 Vaccine Rejection-An International Overview. Vaccines, 2021, 9(1), 44.

Kachurka, R.; Krawczyk, M.W.; Rachubik, J. Persuasive messages will not raise COVID-19 vaccine acceptance. Evidence from nation-wide online experiment. Work. Papers. Univ. Wars. Fac. Econ. Sci. 2021, 7, 335.

Sowa, P.; Kiszkiel, Ł.; Laskowski, P.P.; Alimowski, M.; Szczerbiński, Ł.; Paniczko, M.; Moniuszko-Malinowska, A.; Kamiński, K. COVID-19 Vaccine Hesitancy in Poland-Multifactorial Impact Trajectories. Vaccines (Basel) 2021, 9(8), 876.

Is it because the contents of the intervention hasn’t been focused on vaccination?

Regarding the Reviewer’s comment, we addressed this in the text as follows:

In September 2020, when the educational intervention was conducted, there was no COVID-19 vaccine available for the general population. Therefore, the presentation focused on those vaccine candidates which were under investigation regarding various phases of randomized clinical trials. Additionally, the need for a COVID-19 vaccination was also discussed during the presentation, as well as the benefits of vaccination on the individual and the general population.

Has the peer-based educational intervention tried to overcome misinformation?

Yes, our peer-based educational intervention attempted to overcome misinformation. Regarding the Reviewer’s comment we addressed this issue in the Methods section as follows:

The objectives of the COVID-19 teaching were:

  1. To increase general knowledge about SARS-Cov-2/COVID-19.
  1. To clarify transmission modes, risk factors and preventive methods.
  2. To reduce existing myths and unconfirmed information regarding SARS-Cov-2 infection.
  3. To increase SARS-Cov-2 awareness and improve school student attitudes towards SARS-Cov-2/COVID-19, including the willingness to vaccinate.

Is it because peers may not particularly encourage vaccination because peers themselves are reluctant to get COVID-19 shot?

We do not believe this was the reason why the current peer-based educational intervention did not play strong enough role regarding this particular issue. Medical students who were conducting the educational intervention undoubtedly stated that by joining educational intervention in high schools, they improved their own SARS-Cov-2 knowledge. Furthermore, this inspired them to be vaccinated against COVID-19 (in spring 2021) and to actively join an online National Health Fund campaign (in June 2021) promoting COVID-19 immunization among young people.

The link to the web based information film, which has already had 1578 views, is below:

(https://www.nfz-zielonagora.pl/PL/1048/7706/Nie_ma_mutacji_bez_replikacji__Nie_replikuj__Zaszczep_sie/)

We addressed the Reviewer’s comment as follows:

Our junior colleagues undoubtedly stated that by joining and taking part in an educational intervention in high schools, they improved their own SARS-Cov-2 knowledge. Furthermore, this inspired them to be vaccinated against COVID-19 (in spring 2021) and to actively join an online National Health Fund campaign (in June 2021) promoting COVID-19 immunization among young people.

Reviewer 2 Report

See attached

Reviewer 3 Report

This is an interesting and well written paper. It´s main merit is that it describes an intervention using a design that easily can be implemented on a grand scale.

However, I have some questions about the method which is not clearly stated in the manuscript. Information about the selection process for the student peer-educators is missing. Which factors were decisive when choosing the medical students? Were they randomly assigned to the chosen schools or must certain conditions be met?

Author Response

We would like to thank the Reviewer for this valuable comment. We amended the text according to the Reviewer’s suggestions as follows:

In March 2020, the University of Zielona Gora Medical Faculty students who were the best in their class, including the highest grades obtained during an Infectious Diseases course, were invited to join the Student Research Group. This was led by an academic professor specializing in infectious diseases and epidemiology. During the following months their teaching skills, including a listener-centered approach and effective communication, as well as personal qualities, such as interpersonal relationships and enthusiasm for teaching – all essential to effective role modeling - were examined. Finally, six 5th year medical students, 3 females and 3 males aged 22-24 years, were chosen to conduct peer-based educational interventions at the selected high schools.

Due to COVID-19 pandemic travel restrictions, students were assigned to provinces and schools which were located closest to their places of residence.

Round 2

Reviewer 3 Report

The manuscript in this revised form thoroughly describes the methodology of the study and what conclusions can be drawn from the analyses. The paragraphs that have been included in the discussion section also add valuable information to the manuscript.